# Study of the Influence of Non-Genetic Factors on the Growth and Development Traits and Cashmere Production Traits of Inner Mongolia White Cashmere Goats (Erlangshan Type)

**DOI:** 10.3390/vetsci11070308

**Published:** 2024-07-10

**Authors:** Yue Shi, Yunpeng Qi, Yan Liu, Youjun Rong, Xiaofang Ao, Mingzhu Zhang, Qincheng Xia, Yanjun Zhang, Ruijun Wang

**Affiliations:** 1College of Animal Science, Inner Mongolia Agricultural University, Hohhot 010018, China; yy13847436106@163.com (Y.S.);; 2Erlangshan Ranch of Inner Mongolia Beiping Textile Co., Ltd., Bayan Nur 015000, China; yunpengqi666@163.com; 3College of Vocational and Technical, Inner Mongolia Agricultural University, Baotou 014109, China; 4Key Laboratory of Mutton Sheep Genetics and Breeding, Ministry of Agriculture, Hohhot 010018, China; 5Key Laboratory of Goat and Sheep Genetics, Breeding and Reproduction in Inner Mongolia Autonomous Region, Hohhot 010018, China

**Keywords:** non-genetic factors, growth trait, cashmere trait, body size, Inner Mongolia white cashmere goats (Erlangshan type)

## Abstract

**Simple Summary:**

The Inner Mongolia white Cashmere goat is a breed of goat formed by long-term breeding that is used for mutton and cashmere and can be divided into the Arbas type, Erlangshan type, and Alashan type. The growth and development traits of livestock are affected by many factors, which can be roughly divided into two categories: genetic factors and non-genetic factors. Non-genetic factors generally include sex, birth month, birth type, birth year, group, and some other environmental factors and their interaction effects. To study their effects on different traits is to analyze the reasons for the differences in livestock production performance and put forward major suggestions so as to provide a theoretical basis and reference for the selection breeding of livestock and improve the economic benefits of livestock breeding. Analysis of variance (ANOVA) is a collection method used to compare multiple mean values of different groups. This study analyzed the influence of non-genetic factors on the growth, development, and cashmere production performance traits of Inner Mongolia white cashmere goats (Erlangshan type) and finally identified the non-genetic factors affecting growth, development, and cashmere production traits of Inner Mongolia white cashmere goats (Erlangshan type).

**Abstract:**

The purpose of this study was to investigate the effects of non-genetic factors on the growth and development performance of Inner Mongolia white cashmere goats (Erlanghan type), such as birth weight (BW), weaning weight (WW), 6-month weight (6 WT), 12-month weight (12 WT), body height (BH), and body length (BL), and wool production performance, such as cashmere fineness (CF), cashmere thickness (CT), and cashmere yield (CY). The research objects were 4654 kids produced by 45 buck goats and 2269 doe goats in the Erlang Mountain Ranch of Beiping Textile Co., Ltd., Inner Mongolia, from 2020 to 2023. Based on the generalized linear model, ANOVA was used to analyze the effects of non-genetic factors, such as birth year (Y), birth month (M), sex (S), birth type (T), birth herd (H), assay flock (F), age at measurement (MA), and the age of doe goats at lambing (DLA), on growth and development traits and cashmere traits. The results show that the birth weight (BW), weaning weight (WW), 6-month weight (6 WT), 12-month weight (12 WT), body length (BL), body height (BH), chest depth (CD), chest width (CW), chest circumference (CC), cannon circumference (CNC), wool length (WL), and cashmere yield (CY) of buck goats were significantly higher than those of doe goats (*p* < 0.01), and the fineness of the cashmere produced by doe goats was significantly finer than that produced by buck goats (*p* < 0.01). The birth weight, weaning weight, and 6-month weight of single kids were significantly higher than those of multiple kids (*p* < 0.01), but the effect on the 12-month weight was not significant (*p* > 0.05). The age of doe goats at lambing had significant effects on birth weight, weaning weight, and 6-month weight (*p* < 0.01). Assay flock and age at measurement had significant effects on cashmere fineness, cashmere thickness, and cashmere yield (*p* < 0.01). This study will provide a basis for the scientific breeding and management of cashmere goats and lay a foundation for the setting of fixed effects in the genetic evaluation model of Inner Mongolia white cashmere goats (Erlangshan type).

## 1. Introduction

With the development of the rural economy, the structure of the agricultural industry has adjusted accordingly, and goat farming has become an important support for the development of this economy [1]. The plant species there are typically resistant to drought, cold temperatures, salinity, and alkalinity [2]. With their unique geographical advantages and strong survival ability, cashmere goats have developed rapidly and expanded. Not only have they promoted the development of China’s cashmere goat industry and increased farmers’ income, but they have also boosted China’s economy. After decades of breeding of this variety, its comprehensive performance has reached the national advanced level. In the Inner Mongolia Autonomous Region, the scale of the farming of Inner Mongolia Erlangshan white cashmere goats (the abbreviation IMEWCG will be used in the following parts) has expanded, as has the production of the excellent national cashmere meat combined-type varieties. The production performance of cashmere goats has also improved significantly because of the adjustment in breeding techniques used by farmers. The meat production potential of the IMEWCG has continuously been explored and developed. Thus, they are famous for not only cashmere production performance but also meat production performance. IMEWCGs are distributed in the desert steppe and half desert steppe of western Inner Mongolia. The vegetation there is dominated by leguminous grass and shrubs with a high protein content and a large number of herbal ingredients. This makes the IMEWCG exhibit the following traits: a large physique, high prolificacy, long fibers, high cashmere yield. Additionally, it has the advantages of drought-resistant capacity, cold tolerance, and roughage resistance. At the same time, it has some other advantages, such as low incidence, strong disease resistance, and excellent adaptability [3]. The cashmere produced by the Inner Mongolia Arbas Cashmere Goat has the characteristics of brightness, color, elasticity, fine diameter, and softness [4]. At present, the finest cashmere fiber from the Inner Mongolia Arbas Cashmere goat is 14.2 microns, but the major fineness is 15 microns. Average cashmere production is more than 1200 g, and some outstanding adult buck goats produce 2300 g and adult doe goats produce 2000 g. The average thickness has reached 7.2 cm, and the net cashmere rate has reached 70%. This cashmere exhibits excellent luster, which is why it is known as the “natural fiber gem”.

Not only do the IMEWCGs exhibit excellent cashmere production performance, but the meat product also holds a key position in the international meat trade. This meat is a rare and precious food ingredient that depends on feeding in free-range natural pastures with wild ground pepper, sand onion, and dozens of other Chinese herbs. The meat is deliciously fresh and fatty but not greasy. In particular, the meat has the advantages of less fat, more lean meat, light odor, and easy digestibility, and it is a delicious ingredient, so it is an ingredient that consumers like. A piece of freshly slaughtered mutton is truly delicious, exuding a rich aroma, and its texture is tender and juicy, with a smooth and velvety feel.

There are many factors that could affect the traits of cashmere and meat production, which can be broadly categorized into genetic and non-genetic factors.

This is the basis for analyzing the heritability of cashmere traits and meat production performance, as well as the criteria for breeding methods to estimate genetic parameters accurately.

However, non-genetic factors should be corrected when estimating the genetic parameters for production traits. Thus, the first step of IMEWCG breeding is to study the influence of non-genetic factors on the production performance of Inner Mongolia white cashmere goats [5]. Non-genetic factors include sex, age, growth site, etc.

Early growth traits include birth weight and weaning weight, which can accelerate the progress of breeding in the early selection of breeding goats. Not only can the birth weight of kids reflect the growth and development situation, but it can also be an important reference index for whether the goat can be retained for breeding [6,7]. The weaning weight of kids was significantly correlated with their sex and birth type and the age of doe goats at lambing [8]. For breeding animals, fecundity is productivity, and increasing the weaning weight of kids can significantly shorten their breeding time and improve the accuracy of selection. Growth and development traits refer to the weight and body size characteristics in different age stages, which are important indicators reflecting the production capacity of meat goats. Analysis of the phenotypic measurement data of body weight and body size in different growth stages is helpful in predicting and evaluating the meat production capacity of different meat goat breeds, and it is also helpful in rationally formulating reference standards for the growth and development of meat goats in different stages [9].

Multiple studies have shown that the growth performance and cashmere production performance of other types of Inner Mongolia white cashmere goats may be influenced by non-genetic factors such as location, nutritional level, and climate [10,11,12], but it has not been reported in the Erlangshan type.

Therefore, taking full account of the influence of genetic and non-genetic factors can enhance the effectiveness of breeding and improvement, facilitate the development of reasonable breeding programs, and improve the accuracy of selection.

Non-genetic factors related to the performance of the IMEWCG need to be explored to improve the structure of the IMEWCG group and improve the performance of meat production to match the excellent performance of cashmere production. Thus, it can lay the foundation for further adjustments to current breeding programs, thereby enhancing the economic benefits for farmers.

This study aimed to determine the influence of non-genetic factors, including birth year, birth month, birth type, birth herd, sex, assay flock, age of doe goats at lambing, age at measurement of growth, and cashmere production traits of the IMEWCG.

## 2. Materials and Methods

### 2.1. Data Collection and Studied Traits

The data and pedigree information of this study were obtained from Erlang Ranch of Beiping Textile Co., Ltd., Inner Mongolia.

First, the original data of the growth and development traits and cashmere traits of the IMEWCGs from 2020 to 2023 were compiled according to the following requirements: individuals with blurred or incorrect ear numbers, with no sex, and with no production performance identification records were removed, and information such as incomplete phenotype records was eliminated.

Ultimately, 4654 kid records originating from 45 buck goats and 2269 doe goats were screened from the database.

The growth and development traits include the birth weight of the kids (BW), weaning weight (WW), weight at 6 months (6 WT), and weight at 12 months (12 WT); cashmere traits include cashmere thickness (CT), cashmere fineness (CF), wool length (WL), and cashmere yield (CY); and body size traits at one year of age include body length (BL), body height (BH), chest depth (CD), chest width (CW), chest circumference (CC), and cannon circumference (CNC). Descriptive statistics regarding phenotypic evaluation values of the IMEWCG are shown in Table 1.

### 2.2. Raising and Management

The experimental farm, located in Urad Middle Banner, is situated in the western part of the Inner Mongolia Autonomous Region and the northeastern part of Bayannur City, between 41°07′ and 41°28′ north latitude and 107°16′ and 109°42′ east longitude. It is located in the western part of the Inner Mongolia Plateau, belonging to the semi-desert zone with desert steppes as the main landscape.

The natural grassland vegetation there is sparse and low, dominated by perennial shrubs and tussock grasses. It is a preserved ranch for the IMEWCGs designated by the autonomous region.

Kids were weighed at birth for their birth weight and, at the same time, marked with ear numbers.

From birth to 30 days of age, the kids were raised in captivity, and from this time, they grew and developed rapidly, but the nutrition in breast milk gradually decreased, so it was difficult to maintain the nutritional supply to the goat kids. From 30 days old to weaning, kids were nursed twice a day, morning and evening, for two hours each time. During the daytime, the kids followed the doe goat herds to graze and forage, while at night, they were kept in the same pen as the doe goats. The kid herds adopted a unified weaning management model supplemented with alfalfa green hay. The kids were weaned at around 90 days of age and were weighed at that time. After weaning, the kids were transferred to a breeding group for all-day grazing, divided into the buck and doe flocks. Body weight was measured at 6 and 12 months. The cashmere thickness, wool length, and body traits of yearling goats were measured in March every year; the cashmere yield of all flocks was measured in April to May, and the weight after fleecing was measured in June.

All body weights were measured after fasting 12 h. The areas for measuring wool length and cashmere thickness were located 10 cm behind the shoulder blade on the side of the body. After straightening the wool, the natural length of the hair length and the wool thickness were measured with a ruler. The cashmere fineness was about 20 g of cashmere at the same position and was determined by OFDA 2000 in the laboratory. The cashmere yield is the total fleece weight after the fleece fibers are carded off from the goat body, without the coarse wool.

### 2.3. Statistical Analyses

This study used the ANOVA function in R to determine whether non-genetic factors would affect the study traits, and the differences between different levels of the same significant factor were compared using Duncan’s test and Tukey’s test. The fitted model was as follows:

Model 1: this was used to analyze the non-genetic factors affecting the birth weight and weaning weight of the IMEWCGs:$$Y_{hijklmn}=\mu +H_{h}+Y_{i}+S_{j}+T_{k}+ELA_{l}+M_{m}+e_{hjjklmn}$$
where $Y_{hijklmn}$ is the phenotypic value of the trait, *μ* refers to the overall mean, H refers to the herd, $Y_{i}$ refers to the year of birth, $S_{j}$ refers to the sex of the goat, $T_{k}$ refers to the birth type of the goat, $ELA_{l}$ refers to the age of doe goats at lambing, $M_{m}$ refers to the birth year of kids, and $e_{hjjklmn}$ is a random error. 

Model 2: this was used to analyze the non-genetic factors affecting the 6-month weight of the IMEWCGs:$$Y_{jkln}=\mu +S_{j}+T_{k}+ELA_{l}+e_{jkln}$$
where $Y_{jkln}$ is the phenotypic value of traits, *μ* refers to the overall mean, $S_{j}$ refers to the sex of the goat, $T_{k}$ refers to the birth type of the goat, $ELA_{l}$ refers to the age of doe goats at lambing, and $e_{jkln}$ is a random error.

Model 3: this was used to analyze the non-genetic factors affecting the 12-month weight of the IMEWCGs:$$Y_{jkon}=\mu +S_{j}+T_{k}+F_{o}+e_{jkon}$$
where $Y_{jkon}$ is the phenotypic value of traits, *μ* refers to the overall mean, $S_{j}$ refers to the sex of the goat, $T_{k}$ refers to the birth type of the goat, $F_{o}$ refers to the assay flock of the goat, and $e_{jkon}$ is a random error.

Model 4: this was used to analyze the non-genetic factors affecting the body size of the IMEWCGs:$$Y_{jon}=\mu +S_{j}+F_{o}+e_{jon}$$
where $Y_{jon}$ is the phenotypic value of traits, *μ* refers to the overall mean, $S_{j}$ refers to the sex of the goat, $F_{o}$ refers to the assay flock of the goat, and $e_{jon}$ is a random error.

Model 5: this was used to analyze the non-genetic factors affecting the cashmere production performance of the IMEWCGs:$$Y_{jopn}=\mu +S_{j}+F_{o}+MA_{p}+e_{jopn}$$
where $Y_{jopn}$ is the phenotypic value of traits, *μ* refers to the overall mean, $S_{j}$ refers to the sex of the goat, $F_{o}$ is the herd of feeding, $MA_{p}$ is the age of measurement, and $e_{jopn}$ is a random error.

## 3. Results

### 3.1. Study 1: Exploring the Non-Genetic Factors Affecting the Growth and Development Traits of the IMEWCGs

As shown in Table 1, Table 2, Table 3, Table 4 and Table 5, H, Y, M, S, T, and DLA had significant effects on the birth weight and weaning weight of the IMEWCGs (*p* < 0.01). Both weights of buck kids were significantly higher than those of doe kids. Both weights of singleton kids were significantly higher than those of multiple kids. Both weights of the kids delivered by 3-year-old doe goats were higher than those delivered by doe goats of other ages. Sex had a significant effect on both 6- and 12-month weight, and, buck kids were heavier than doe kids for both weights. The birth type and lambing age of doe goats had a significant effect on the 6-month weight. However, it did not affect the weight at 12 months. The weight of singleton kids was significantly higher than that of multiple kids, and it was significantly lower than those of other doe goats. Sex had no effect on the body length of the IMEWCGs but had a significant effect on body height, chest depth, chest width, chest circumference, and cannon circumference, and all traits of the buck goats were significantly higher than those of the doe goats. The determination flock had a significant effect on body length, chest depth, and chest width but did not affect body height and cannon circumference.

### 3.2. Study 2: Exploring the Non-Genetic Factors Affecting the Cashmere Characteristics of the IMEWCGs

Table 6 shows that the assay flock, measured age, and sex significantly affected the mean fineness of the IMEWCGs (*p* < 0.01). The average fineness of cashmere produced by 3-year-old goats was significantly lower than that produced by 4-year-old goats, and that produced by doe goats was significantly lower than that produced by buck goats. The assay flock, year, and age had significant effects on the fineness of the IMEWCGs (*p* < 0.01). Sex and birth type had no significant effect on fineness. The thickness of cashmere produced by 2-year-old goats was significantly higher than that of 3- and 4-year-old goats. The assay flock, measurement age, and sex had a significant effect on the fineness of the IMEWCGs (*p* < 0.01). The fineness of the cashmere produced by 2-year-old goats was significantly lower than that produced by goats of other ages. The determination year, determination flock, and determination age had no effect on the length of the wool produced by the IMEWCG (*p* > 0.05). However, the length of wool produced by buck goats was significantly higher than that of doe goats. The wool length of single births was significantly longer than that of multiple births. Every factor had a significant effect on cashmere yield (*p* < 0.01). The cashmere yield of goats in group 22 was significantly higher than that of goats in the other groups. The cashmere yield of 2-year-old goats was significantly higher than that of goats of other ages. The cashmere yield of the buck goats was significantly higher than that of the doe goats.

## 4. Discussion

### 4.1. Growth and Development Performance

The findings of this study indicate that birth type, birth year, birth month, sex, age of doe goats at lambing, and birth herd had a significant impact on the birth weight and weaning weight of the IMEWCGs. Siddalingamurthy concluded that the factors affecting the birth weight and weaning weight of Mandya sheep were sex and birth year [13]. Chandran’s findings suggest that the birth type and sex of kids have a significant impact on the birth weight of Shahabadi sheep [14]. Vlahek’s research on Romanov sheep proves that the birth type, sex, birth group, and birth year of kids have extremely significant effects on birth weight [15]. The above results are consistent with the results of this study. Jamshid Ehsaninia’s study on Sangsari sheep [16] concluded that the birth weight and weaning weight of ram lambs were significantly higher than those of ewe lambs, and the birth weight of single births was significantly higher than that of multiple births, which is consistent with the results of this study. The studies of Eskandarinasab [17] in Afshari sheep, Shahdad [18,19] in Kourdi sheep, and Bangar [20] in Deccani sheep showed similar conclusions. Jamshid Ehsaninia’s study of Sangsari sheep and G. S. Dhakad’s study of Malpura sheep [16,21] showed that the lambs of young ewes had a lower birth weight, which is inconsistent with the present study. For 6-month weight, Shahdadi’s study of Kourdi sheep [18] concluded that sex and birth type had a significant effect on 6- and 12-month weights. Shirzeyli’s study of Markhoz goats [22] showed that the 6-month weight and 12-month weight of buck goats were significantly higher than those of the doe goats. Dangi concluded in his study on crossbred sheep that the 12-month weight of ram lambs was significantly greater than those of ewe lambs [23]. This study concluded that sex and birth type had a very significant effect on the weight at 6 months but no effect on the weight at 12 months. Sarma’s study of Assam Hill goats [24] concluded that sex had a significant effect on the body size traits of goats, and Kumar’s study of Nellore sheep [25] came to the same conclusion. Mandal [26] pointed out that the sex and birth type of Muzaffarnagari sheep had a very significant effect on body height, body length, chest circumference, and cannon circumference. This study concluded that sex had a very significant effect on body height, body length, chest circumference, and cannon circumference, but birth type had no effect on those traits. This may be due to different nutrient levels.

During the embryonic period, the distribution of nutrients is uneven. Moreover, during the lactation stage, the milk production capacity of doe goats is limited. This may lead to lower weaning weights in multiple-born kids than in single-born kids. Competitive behavior occurs between multiple births, and each kid receives fewer nutrients in each stage of growth and development, but single births can be fully cared for by the doe goats and receive all the nutrients. Therefore, the nutritional level of multiple births is not as good as that of a single birth, so the birth weight and weaning weight of single births are significantly higher than those of multiple births. After kids are weaned at 3 months of age, the feeding program changes to completely free feeding, fewer and fewer nutrients are supplied by the doe goats, their physical fitness continuously improves, immunity gradually strengthens, and they no longer exhibit differences in their growth and developmental traits between multiple- and single-birth goats.

Kids born in different herds lived with doe goats of different managers and inhabited various conditions; each manager had their own feeding programs and techniques, and different breeders had different personalities, so feeding effects differed. The body function of young doe goats is better than that of mature doe goats, and they have a greater ability to conceive embryos, so the birth weight of kids produced by young doe goats is larger than that of mature doe goats. With the increase in age of the doe goats, the maternal behavior and milk production are improved, so the weaning weight of kids produced by mature doe goats is significantly higher than that of young doe goats. There are differences in climate between different years, which leads to different growths of forage grass, and the birth environment of the kids is also different, so the birth year has a significant impact on various early growth traits. The effects of birth month on domestic animals mainly come from the difference in season and climate. Inner Mongolia is located in the northern frontier of China, with a relatively high latitude and a vast plateau area. It is far away from the ocean and bordered by mountain ranges. The main climate is a temperate continental monsoon, gradually warming up from March, with the temperature remaining stable in April and May. After accumulating nutrients throughout the winter, the doe goats are in good physical condition to give birth to healthy kids. Therefore, kids born in April and May have significantly higher birth weight and weaning weight compared to those born in other months.

Different months and years bring about variations in the natural environment, including temperature, humidity, and precipitation levels, which can affect the growth and development of pasture grasses. Consequently, these differences can have an impact on the early growth characteristics of kids, depending on the specific month and year.

### 4.2. Performance of the Cashmere Fiber

The effect of age on cashmere quality is related to the growth and development of individuals themselves. With the increase in age, the endocrine, digestive, nervous, and other physiological systems of cashmere goats gradually develop and improve, and the utilization efficiency of forage increases significantly, leading to changes in the quality of the cashmere [27]. Determination year and herd effects are intricate comprehensive effects that integrate natural ecological conditions, feeding management, and herd status. Variations in rainfall levels across different years lead to differences in forage quality, and the supply of supplementary feed also has an impact on the production performance of individuals. Different groups are responsible for different staff, and the feeding and management mode will lead to differences in the performance of wool and cashmere quality traits in different herds.

As animals age, their bodily development gradually changes. During the gestation and lactation periods, multiple births often compete for nutrients, resulting in fewer nutrients absorbed by each individual. However, after weaning and being separated into different herds, each animal has its independent pasture, and combined with reasonable feeding management, the gap in development gradually narrows.

The results of this study indicate that the year of measurement, the herd at measurement, and the age at measurement have extremely significant effects on average fineness, cashmere thickness, cashmere fineness, and cashmere yield but no effect on hair length. Sex has an extremely significant impact on average fineness, cashmere fineness, hair length, and cashmere yield but no effect on cashmere thickness.

Zhou et al. [28] pointed out that the cashmere fineness, wool length, and cashmere yield of buck goats are significantly higher than those of doe goats, and the development degree of hair follicles also affects the wool yield of buck goats and doe goats. Cashmere is differentiated from secondary hair follicles, and the higher the density of secondary hair follicles, the higher the cashmere yield.

According to Cloete’s study on Merino sheep [29], the wool yield of rams was significantly higher than that of ewes (*p* < 0.01), while the wool fineness of ewes was significantly lower than that of rams, which is consistent with the results of the present study. However, Cimen found that the fiber diameter of Karayaka sheep was not related to sex, which may have something to do with the different varieties studied [30]. With the increase in age, secondary hair follicles continue to develop, their functions constantly improve, and the fleece production capacity is also constantly enhanced, resulting in cashmere fibers becoming longer and coarser but with cashmere production increasing [31]. Toigonbaev [32] found that the cashmere production of indigenous cashmere goats in southern Kyrgyzstan gradually increased before 4 years of age, and buck goats produced more cashmere than doe goats, which is consistent with the results of this study. This is slightly different to the results of this study, which might be related to the test breed, feeding environment, and sample size.

Rashidi’s study of Markhoz goats [33] showed that ram production was significantly higher than that of ewes, from 2 to 4, and cashmere yield increased with the age, which is consistent with our research results. Wang [34] concluded that the production of Inner Mongolia cashmere goats was significantly higher than that of doe goats.

The fineness of cashmere is an important index to measure the quality of cashmere and a key factor in determining the quality of cashmere yarn. In addition, another study concluded that the sex of Inner Mongolia cashmere goats has no significant impact on cashmere fineness [28]. This may be due to regional environmental differences. The fineness of cashmere tends to thicken with age, and Antonini [35] in Alashan cashmere goat showed that 1-year-old goats produce thinner cashmere than 3- and 4-year-old goats. Wang Zhiying’s [36] study of Inner Mongolia cashmere goats showed that fiber diameter increased with age, and the cashmere produced by 2-year-old kids was significantly thinner than that produced by 5-year-old goats, while after 2 years of age, the cashmere length decreased with age. The cashmere fineness gradually increases with age, and all of the above studies showed the same results as this one. 

Because goats do not shed their hair every year, there are relatively few reports about goat wool. Karen’s [37] study showed that age had a significant effect on the hair length of different parts of Zhongwei goats. The studies of Gregor on Angora goats and Wen Hui Li on Alpine Merino sheep both show that age had a very significant effect on fiber length [38,39]. However, this study found that age had no significant effect on wool length, which may be related to the sampling parts and varieties being different.

## 5. Conclusions

In summary, the following conclusions are drawn:Year, month, and group of birth and their reciprocal effects had highly significant effects on the birth weight and weaning weight of the IMEWCGs.Buck kids had better growth and development performance than doe kids, so after weaning, buck goats and doe goats should be kept in separate groups to provide different feeding standards.Birth type had no effect on the one-year old weight and subsequent traits of the IMEWCGs. Therefore, the effects of birth type on various traits should be considered and corrected in early selection.The birth weight and weaning weight of kids from 3-year-old doe goats were larger, so it is recommended to select 2-year-old doe goats as much as possible when breeding.The fineness and thickness of cashmere produced by 4-year-old IMEWCGs were excellent, but the cashmere yield was low, and the doe goats produced less and finer cashmere than the buck goats. Whether there is a correlation between cashmere fineness and cashmere yield needs to be further explored in future studies.

## Figures and Tables

**Table 1 vetsci-11-00308-t001:** Characteristics of the data structure for phenotypic data of Inner Mongolia cashmere goats.

Item	No. of Animals	No. of Sires	No. of Dams	Mean	S.D.	C.V. (%)
BW (kg)	4654	45	2269	2.53	0.51	20.24
WW (kg)	2010	26	1379	17.15	3.47	20.22
6 WT (kg)	835	26	761	20.20	3.28	16.22
12 WT (kg)	769	14	717	29.16	5.61	19.23
CT (cm)	2373	26	1434	5.14	1.16	22.53
CF (μm)	2289	26	1402	15.02	0.99	6.58
WL (cm)	2386	26	1440	14.16	2.73	19.30
CY (g)	2336	26	1243	719.29	296.98	41.29
BH (cm)	756	14	707	55.02	3.82	6.95
BL (cm)	758	14	708	58.26	4.12	7.07
CD (cm)	754	14	705	26.32	2.15	8.15
CW (cm)	752	14	703	17.91	2.42	13.53
CC (cm)	753	14	704	74.54	5.63	7.56
CNC (cm)	755	14	705	8.38	0.57	6.82

**Table 2 vetsci-11-00308-t002:** Least-squares means ± S.D. for birth weight.

Fixed Factors		*n*	$\bar{x}\pm S$ (kg)
H ***	1	477	2.51 ± 0.49 ^c^
	2	183	2.33 ± 0.27 ^de^
	3	191	2.31 ± 0.24 ^e^
	4	977	2.38 ± 0.56 ^d^
	5	783	2.61 ± 0.52 ^b^
	6	725	2.78 ± 0.43 ^a^
	7	1317	2.52 ± 0.5 ^c^
Y ***	1	47	2.7 ± 0.37 ^b^
	2	414	2.31 ± 0.46 ^a^
	3	3306	2.51 ± 0.52 ^a^
	4	790	2.71 ± 0.46 ^b^
	5	96	2.66 ± 0.47 ^b^
M ***	1	47	2.7 ± 0.37
	2	414	2.31 ± 0.46 ^a^
	3	3306	2.51 ± 0.52 ^a^
	4	790	2.71 ± 0.46 ^b^
	5	96	2.66 ± 0.47 ^b^
S ***	M	2344	2.64 ± 0.52 ^a^
	F	2304	2.42 ± 0.48 ^b^
T ***	1	3838	2.61 ± 0.48 ^a^
	2	835	2.15 ± 0.47 ^b^
DLA ***	2	1051	2.52 ± 0.59 ^b^
	3	1361	2.60 ± 0.51 ^a^
	4	1113	2.46 ± 0.47 ^c^
	5	11,228	2.51 ± 0.45 ^b^
H*Y*M		***	

Note: H, birth herd; Y, birth year; M, birth month; S, sex; T, birth type; DLA, age of doe goats at lambing; H*Y*M, birth herd, birth year, and birth month and their interaction effects. Averages with different letters within the same column are significantly different from each other. *** (*p* < 0.01). NS: non-significant. a, b, c, d and e indicate the significance of differences between groups.

**Table 3 vetsci-11-00308-t003:** Least-squares means ± S.D. for weaning weight.

Fixed Factor		*n*	$\bar{x}\pm S$ (kg)
H ***	1	317	15.86 ± 2.94 ^c^
	4	450	18.02 ± 3.33 ^a^
	5	297	16.03 ± 3.55 ^c^
	6	402	18.13 ± 3.16 ^a^
	7	543	17.12 ± 3.44 ^b^
Y ***	2020	54	14.86 ± 2.61 ^b^
	2021	1078	17.49 ± 3.62 ^a^
	2022	877	16.9 ± 3.12 ^a^
M ***	2	234	17.35 ± 3.39 ^a^
	3	1492	17.78 ± 3.16 ^a^
	4	241	14.08 ± 2.66 ^b^
	5	42	11.95 ± 2.46 ^c^
S ***	M	902	17.95 ± 3.44 ^a^
	F	1102	16.52 ± 3.27 ^b^
T ***	1	1465	17.43 ± 3.41 ^a^
	2	544	16.44 ± 3.34 ^b^
DLA ***	2	70	15.56 ± 3.13 ^b^
	3	1281	17.21 ± 3.55 ^a^
	4	658	17.24 ± 3.15 ^a^
H*Y*M		***	

Note: H, birth herd; Y, birth year; M, birth month; S, sex; T, birth type; DLA, age of doe goats at lambing; H*Y*M, birth herd, birth year, and birth month and their interaction effects. Averages with different letters within the same column are significantly different from each other. *** (*p* < 0.01). NS: non-significant. a, b, and c indicate the significance of differences between groups.

**Table 4 vetsci-11-00308-t004:** Least-squares means ± S.D. for 6 MWT and 12 MWT.

Trait	Fixed Factor		*n*	$\bar{x}\pm S$ (kg)
6 WT	S	M	359	20.38 ± 3.34 ^a^
	***	F	349	19.63 ± 2.53 ^b^
	T	1	615	21.58 ± 2.88 ^a^
	***	2	93	18.79 ± 3.33 ^b^
	DLA	3	138	19.05 ± 3.03 ^b^
	***	4	172	20.33 ± 3.42 ^a^
		5	398	20.55 ± 3.22 ^a^
12 WT	S	1	383	32.9 ± 4.41 ^a^
	***	2	384	25.34 ± 3.68 ^b^
	T	1	673	29.12 ± 5.58
	NS	2	96	29.06 ± 5.27
	F	11	376	28.92 ± 5.64
	NS	21	145	28.98 ± 5.53
		22	248	29.48 ± 5.41

Note: S, sex; T, birth type; DLA, age of doe goats at lambing; F, assay flock. Averages with different letters within the same column are significantly different from each other. *** (*p* < 0.01). NS: non-significant. a, b, and indicate the significance of differences between groups.

**Table 5 vetsci-11-00308-t005:** Least-squares means ± S.D. for body measurement traits.

Fixed Factors	BH (cm)		BL (cm)		CD (cm)		CW (cm)		CC (cm)		AC (cm)	
	*n*	$\bar{x}\pm S$	*n*	$\bar{x}\pm S$	*n*	$\bar{x}\pm S$	*n*	$\bar{x}\pm S$	*n*	$\bar{x}\pm S$	*n*	$\bar{x}\pm S$
S	***		***		***		***		***		NS	
M	369	56.92 ± 3.20 ^a^	270	59.99 ± 3.51 ^a^	370	27.62 ± 1.35 ^a^	368	18.21 ± 1.78 ^a^	368	77.52 ± 4.18 ^a^	367	8.5 ± 0.48 ^a^
F	384	53.33 ± 2.79 ^b^	384	56.57 ± 3.97 ^b^	381	25.11 ± 1.96 ^b^	381	17.5 ± 2.24 ^b^	382	71.64 ± 4.58 ^b^	384	8.24 ± 0.6 ^b^
F	NS		***		***		***		***		***	
14	361	55.1 ± 3.43	362	56.38 ± 3.92 ^c^	359	24.95 ± 1.89 ^c^	359	17.47 ± 2.27 ^b^	360	71.35 ± 4.5 ^c^	362	8.24 ± 0.61 ^b^
21	146	54.52 ± 3.57	146	62.01 ± 3.17 ^a^	146	28.32 ± 1.31 ^a^	145	19.19 ± 1.58 ^a^	144	79.8 ± 3.89 ^a^	145	8.49 ± 0.49 ^a^
22	249	55.39 ± 3.51	249	58.81 ± 3.12 ^b^	249	27.2 ± 1.19 ^b^	248	17.61 ± 1.59 ^b^	249	76.08 ± 3.66 ^b^	247	8.49 ± 0.47 ^a^

Note: S, sex; F, assay flock. Averages with different letters within the same column are significantly different from each other. *** (*p* < 0.01). NS: non-significant. a, b, and c indicate the significance of differences between groups.

**Table 6 vetsci-11-00308-t006:** Least-squares means ± S.D. for cashmere characteristics.

Fixed Factors		CF (μm)		CT (cm)		WL (cm)		CY (g)	
		*n*	$\bar{x}\pm S$	*n*	$\bar{x}\pm S$	*n*	$\bar{x}\pm S$	*n*	$\bar{x}\pm S$
		***		***		NS		***	
F	12	18	15.75 ± 1 ^a^	4	4.83 ± 1.44 ^ab^	4	12.67 ± 3.51	4	476.25 ± 79.83 ^c^
	13	16	15.41 ± 0.73 ^ab^	1	—	2	14 ± 1.41	2	532.5 ± 53.03 ^c^
	14	1144	15.21 ± 0.96 ^abc^	1099	5.12 ± 1.23 ^ab^	1110	14.18 ± 2.74	1097	659.85 ± 221.82 ^c^
	15	29	15.3 ± 0.75 ^abc^	2	5.5 ± 0.71 ^a^	2	16.45 ± 0.64	2	430 ± 253.29 ^c^
	21	704	14.73 ± 0.95 ^c^	787	4.96 ± 0.82 ^ab^	785	14.07 ± 2.72	776	750.36 ± 249.32 ^b^
	22	378	14.91 ± 1.01 ^bc^	479	5.33 ± 1.08 ^a^	438	14.3 ± 2.54	455	788.77 ± 275.74 ^a^
		***		***		NS		***	
MA	2	1540	14.77 ± 0.94 ^c^	1613	5.23 ± 1.13 ^a^	1629	14.17 ± 2.63	1593	756.17 ± 259.33 ^a^
	3	726	15.52 ± 0.88 ^b^	734	4.84 ± 0.94 ^b^	732	14.16 ± 2.82	719	626.05 ± 195.31 ^b^
	4	23	16.04 ± 0.89 ^a^	25	4.96 ± 0.79 ^ab^	25	14.14 ± 3.34	24	598.54 ± 172.09 ^b^
		***		NS		**		***	
S	1	1148	15.21 ± 0.98 ^a^	1211	5.1 ± 0.95	1211	14.31 ± 2.56 ^a^	1169	762.01 ± 257.89 ^a^
	2	1136	14.83 ± 0.96 ^b^	1156	5.11 ± 1.22	1170	14.02 ± 2.82 ^b^	1164	667.19 ± 228.67 ^b^

Note: S, sex; F, assay flock; MA, age of measurement. Averages with different letters within the same column are significantly different from each other. *** (*p* < 0.01). ** (*p* < 0.05). NS: non-significant; a, b, and c indicate the significance of differences between groups.

## Data Availability

The supporting data of this study are available from the corresponding authors upon request.

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
