# Peer review of "Study of the Influence of Non-Genetic Factors on the Growth and Development Traits and Cashmere Production Traits of Inner Mongolia White Cashmere Goats (Erlangshan Type)"

_vetsci, 2024, doi:10.3390/vetsci11070308_

Round 1
Reviewer 1 Report
Comments and Suggestions for Authors
Inform reviewer 1
The present work intended to study the influence of non-genetic factors on growth, development and cashmere production characteristics of Inner Mongolia White cashmere goats, based on the generalized linear model. These non-genetic factors where birth year , birth month , sex, birth type, birth herd , herd of feeding, age of measurement and the age of ewes lambing. Several significant effects on the considered productive characters were found. This work seems to be a continuation of the usual line of work on the characterization of this goat breed done by this group.
General comment: Usually, a male goat is referred to as a "buck" goat. Female goats are called "does" goats. Baby goats are referred to as “kids”. These terms do not appear in the entire work; the usual names in sheep are used.
Abstract: Several sentences in abstract are difficult to understand, please rewrite them:
Lines 28-29: However, the effect on the age was not significant (p>0.05). What factor seems to have no effect on age?
Lines 29-30: The age of ewe lambing had significant effects on birth weight, weaning weight, and 6-month weight when the lambs were borne(p<0.01). What do you mean by “when the lambs were borne”?
Introduction: Very long and messy introduction. “So on” or “etc” are overused; these terms must be specified. Sometimes, verbs and punctuation marks are missing:
Lines 45-46: After the improvement of crossbreeding and seed selection and breeding by scientists and herdsmen for decades. Comprehensive performance has reached the national advanced level. Maybe the period between these two sentences should be removed.
Line 67: product also holds a key position in the international meat trade The meat is a rare and …. Perhaps a period should be included between these two sentences
Several sentences need an explanation:
Lines 57-58: At the same time, it has some other advantages, such as low incidence, strong disease resistance,…. Please explain what disease “low incidence” refers to.
Line 64: The average thickness reached 7.2 cm. Are you sure that these length units (cm) are appropriate?
Material and methods: Several punctuation marks are wrong:
Lines 119-120: as incomplete phenotype records, : this comma should be a period.
The units must be included either in the text or in the table1.
Line 170: What does “gaot” means? Do you want to mean “goat”?
Many mathematical models are show. When is the reason for this? The use of each of them in this study must be explained in detail.
Results: Tables are essentials for understanding results, please improve them.
Tables 2-5 are not referred in the text. Also, they need detailed explanations. Table headers are very uninformative; explain abbreviation, signs like *** and superscripts. Does T*Y*M refer to interactions?
Lines 203-204: The weight of singleton lambs was significantly higher than that of multiple lambs, and it was significantly lower than that of other ewes. Please explain what “other ewes” refer to.
Line 225: What does “flesize” mean?
Discussion: Most of references are about sheep, but the present work is about goat. On the other hand, most of described effects are well known from long time ago. Long paragraphs lack references (lines 264-304; 317-320).
Lines 297-300: Therefore, lambs born in April and May have significantly higher birth weights and weaning weights compared to those born in other months. Several explanations about seasonality of deliveries for these goats are needed.
Lines 307-308: long velvet enzyme system. Please explain.
References:
Most of the references are recent; 20/40 (50%) from the last 10 years and 10/40; (25%) from the las 5 years. However, several references seem incomplete: in some references, both the journal number and pages are missing.

Comments on the Quality of English Language
Moderate editing of English language required
General comment: Usually, a male goat is referred to as a "buck" goat. Female goats are called "does" goats. Baby goats are referred to as “kids”. These terms do not appear in the entire work; the usual names in sheep are used.
In introduction “So on” or “etc” are overused; these terms must be specified. Sometimes, verbs and punctuation marks are missing:
Lines 45-46: After the improvement of crossbreeding and seed selection and breeding by scientists and herdsmen for decades. Comprehensive performance has reached the national advanced level. Maybe the period between these two sentences should be removed.
Line 67: product also holds a key position in the international meat trade The meat is a rare and …. Perhaps a period should be included between these two sentences
Material and methods: Several punctuation marks are wrong:
Lines 119-120: as incomplete phenotype records, : this comma should be a period.
Line 170: What does “gaot” means? Do you want to mean “goat”?
Line 225: What does “flesize” mean?
Author Response
Dear Reviewer:
Thank you for your letter and for the reviewers’ comments concerning our manuscript entitled “Study on the Influence of Non-Genetic Factors on Growth and Development Characters and Cashmere Production Characters of Inner Mongolia White Cashmere Goats (Erlangshan Type)” (vetsci-3054881).Those comments are all valuable and very helpful for revising and improving our paper, as well as the important guiding significance to our researches. We have studied comments carefully and have made correction which we hope meet with approval.
- The male goat has been modified to "buckgoat", female goat has been modified to “dose goat”, and the baby goat has been modified to “kids” in the entire work.
- “So on” or “etc” has been changed by verbs and punctuation marks.
- Lines 45-46: The period between these two sentences,“After the improvement of crossbreeding and seed selection and breeding by scientists and herdsmen for decades. Comprehensive performance has reached the national advanced level.”,has been
- Line 67: The period has been included between “product also holds a key position in the international meat trade”and “The meat is a rare and …. ”
- The punctuation markshave corrected from material and methods
- Lines 119-120: The centence “as incomplete phenotype records,”has been modified to another expression.
- Line 170: This was caused by a spelling error, which has been changed .
- Line 225: It was an incorrect expression and has since been removed.
Thank you and best regards.
Yours sincerely,
Ruijun Wang
Inner Mongolia Agricultural University
306 Zhaowuda Road, Saihan District, Hohhot, P.R.China, 010018
Mobile: 086-15648116152
E-mail: imauwrj@163.com

Reviewer 2 Report
Comments and Suggestions for Authors
The MS entitled “Study on the Influence of Non-Genetic Factors …” written by Shi et al compared the non-genetic factors (birth year, month, type, herd, feeding, sex, measurement, lambing age) on the traits of growth, development, and cashmere production (the weight of birth, weaning,6-month and12-month, body height and length, cashmere fineness, thickness and yield) in Inner Mongolia White Cashmere Goats, which will provide a basis for the scientific breeding and management of cashmere goats and lay a foundation for the setting of fixed effects in the genetic evaluation model of Inner Mongolia white cashmere goats.
General Comments
The studies reported in this manuscript were conducted in an appropriate fashion with objectives being addressed using appropriate study designs and methods to achieve the objectives. The written quality of the manuscript, however, can be markedly improved and the authors should endeavor to do so if this manuscript is to be further considered for publication in Vet. Sci..
Specific Comments
L2: The title:
Growth and 2 Development Characters and Cashmere Production Characters
the traits of Growth, Development, and Cashmere Production
Abstract:
L28: However, the effect on the age was not significant(p>0.05) not clear,what affect the age?
L30: Individual age has a significant effect on cash-30 mere fineness, cashmere thickness, and cashmere yield. Need a p-value
Introduction:
L45: After the improvement of crossbreeding and seed selection and breeding by scientists and 45 herdsmen for decades. Not complete
L48: Inner Mongolia Erlangshan white cashmere goats (IMEWCG), use the abbreviation at the following part
L61: Pls cite the references for the cashmere traits, the same as the meat production.
L75: need references to support the factors affecting other species, and combined the following paragraph
L141: In the Raising and Management part, concise the feeding and add the feed composition if accessible. Pls list how to detect the traits, respectively.
Results:
Tables:The form should be self-explanatory, pls add the notes for the table, explain the abbreviations and symbols in the table, and add the p-value in the text
Discussion: just list some of them
L263: This study concluded that gender had a very significant effect on 263 the traits, but no effect on birth type. The vagueness of this sentence is liable to cause ambiguity
L266: so they consume more nutrients than the female lamb and the digestive capacity of the male lamb and the female lamb is different. Authors did not detect the nutrients data, so it’s not an appropriate explanation. Any references can be cited?
Comments on the Quality of English LanguageExtensive editing of English language required.
Author Response
Dear Reviewers:
Thank you for your letter and for the comments concerning our manuscript entitled “Study on the Influence of Non-Genetic Factors on Growth and Development Characters and Cashmere Production Characters of Inner Mongolia White Cashmere Goats (Erlangshan Type)” (ID: vetsci-3054881). Those comments are all valuable and very helpful for revising and improving our paper, as well as the important guiding significance to our researches.
Responds to the reviewer’s comments:
- L28: “However, the effect on the age was not significant(p>0.05)”, this sentence follows the previous sentence, correctly referring to “the effect of birth type on the 12-month weight was not significan”.
- L30: The p-value has been added to:Individual age has a significant effect on cash-30 mere fineness, cashmere thickness, and cashmere yield.”
- L45:The sentence“After the improvement of crossbreeding and seed selection and breeding by scientists and 45 herdsmen for decades.” and the next sentence “Comprehensive performance has reached the national advanced level.” are a complete sentence , I have changed the comma to a period.
- The“Inner Mongolia Erlangshan white cashmere goats” , has been used the abbreviation “IMEWCG” at the following part.
- L61: Relevant references have been added.
- L75: The information on other species is in the following paragraph.
- L141: Supplementary ingredients have been added, and testing methods for traits have also been added.
- All tables have been annotated to explain abbreviations and symbols that appear in the tables, and p-values have been added to the text.
- L263: This sentence has been deleted because of an error in expression.
Thank you and best regards.
Yours sincerely,
Ruijun Wang
Inner Mongolia Agricultural University
306 Zhaowuda Road, Saihan District, Hohhot, P.R.China, 010018
Mobile: 086-15648116152
E-mail: imauwrj@163.com

Reviewer 3 Report
Comments and Suggestions for Authors
In the study by Shi Y., et al., the researchers attempted to determine the impact of many non-genetic factors on lamb growth characteristics and cashmere production in sheep of Inner Mongolia White Cashmere Goats. In this work, the authors made an extensive comparison of numerous body measurement traits and cashmere characteristics in these sheep. The research was carried out based on data obtained from a large sheep farm over several years. The purpose of the study is justified because it is the first study of this type in sheep of this breed, carried out on such a large scale on numerous research materials, using perfectly refined statistical calculations. I believe that the introduction should be thoroughly edited by a native speaker. The material and methods, results and discussion are unquestionable. For better clarity and understanding of the tables, there should be more descriptions of the table contents under the tables. Moreover, n parameters in tables should be marked with the units in which they are presented (nanograms, grams, micrometers).
All my comments about this work will be detailed in the following paragraphs.
Line 26-27: “than the 26 ram(p<0.01).” - is this written correctly in english?
Line 45-46: After the improvement of crossbreeding and seed selection and breeding by scientists and herdsmen for decades. – semen or sperm not seed rather, and the next sentence - lack of harmony.
Line 57: it has tolerance to cold, drought, and roughage. ??
Line 59-60: so on – it is not a precise scientific term
Line 64: the net cashmere rate reached 70% - what is it?
Line 67: full stop
Line 70: Its flesh layers?
Line 72: so on – it is not a precise scientific term
Line 72-73: “by the majority of consumers” - for carnivorous without heat treatment?
Line 80: comma, space
Line 87: seed selection – semen selection
Line 88: type of birth, specify what this term means
Line 89: fertility is productivity or fertility means productivity ?
Line 97: as well - hard to understand what's going on?
Line 98-99: the sentence should be edited
Line 111: the sentence should be edited to mention about inner Mongolia White Cashmere Goats
Line 129: In what units were the averages (mean) in this and in the remaining tables?
Line 195: all factors? please mention them more clearly
Line 210: what is the unit for n in each parameter? Can you repeat the description of the table below the table to make it more understandable to the reader?
Line 211: same as above
Line 217: birth type - single or twin?
Line 243: Vlahek ? position 15 is Pipan et al.?
Line 250: space
Line 254: there is 19 in brackets, it should be 18
Line 254: birth type? As in Line 217
Line 255: Shirzelyli in 22 reference is Shirzelid?
Line 258: lack of period
Line 260: Sarmal in reference 24 is Sarma
Line 261: just quote the name without the author's name
Line 307: long velvet enzyme system - What does it mean?
Line 307: why from this point RAMS are written in capitals?
Line 332: why Clothe in 29 reference is Swp. C.?
Line 341: space
Line 352: the author's name is enough Antonini
Author Response
Dear Editor,
I would like to thank the respected Reviewers for their constructive comments on our manuscript “Study on the Influence of Non-Genetic Factors on Growth and Development Characters and Cashmere Production Characters of Inner Mongolia White Cashmere Goats (Erlangshan Type)” (ID: vetsci-3054881). I have considered the comments very carefully and have revised the paper accordingly.
- Line 26-27: This sentence has been modified to the correct expressionin English.
- Line 45-46: Inner Mongolia Erlangshan white cashmere goat has not been cross-bred, only this breed breeding, I have revised the original sentence.
- Line 57: What I want to express is that Inner Mongolia Erlangshanwhite cashmere goat has a bit of drought-resistant capacity ,cold-tolerance and resistance roughage, which I have modified in the article.
- Line 59-60: The words “so on”in the article have been revised into a precise scientific term.
- Line 64: The net cashmere ratementioned in the article refers to the percentage of the cashmere hair quality after washing, drying and removing coarse hair and impurities, and using the fixed moisture recovery rate and the fixed fat content.
- Line 67:The sentence has been rewritten and a period has been added.
- Line 70: The word “flesh layers”has been removed.
- Line 72: The word “so on”has been modified to be precise.
- Line 72-73: The original intention of the sentence is to express that consumers tend to use IMEWCG meat as an ingredient, and it is treated with heat, This sentence has been modified.
- Line 80:The comma and space have been added.
- Line 87: What this says is that birth weight can be used to determine whether the goat can be retained for breeding.
- Line 88: The type of birth is what indicates whether a single child or multiple children were born.
- Line 89: The correct expression of this sentence is: fecundity is productivity, and the sentence has been revised.
- Line 97: According to geographical distribution, Inner Mongolia white cashmere goats can be divided into three types: Erlangshan type, Arbas type and Alxa type. This study focuses on Erlansgshan type, and the other types are Arbas type and Alxa type.
- Line 98-99:The sentence has been edited.
- Line 111: The sentence has been edited to mention about inner Mongolia White Cashmere Goats.
- Line 129: The units of the averages (mean) in this and in the remaining tables have been added.
- Line 195: All the factors are listed.
- Line 210: The units in each table have been added.
- Line 217: Birth type is a general term for single and multiple births.hat means both singletons and multiples have no significant effect on fineness.
- Line 243: Redundant references have been removed.
- Line 250: Spaceh has been added.
- Line 254: The 19 in parentheses has been changed to 18.
- Line 254: Birth type is a general term for single and multiple births.That means both singletons and multiples have a significant effect on 6-month weight and 12-month weight.
- Line 255: It is “Shirzelyli”.
- Line 258: It is “lactation periods”.
- Line 260: It’s “Sarma L”in reference 24.
- Line 261: The author's name has been added.
- Line 307: The long velvet enzyme system is an incorrect expression that has been deleted.
- Line 307: It’s a wrong expression, has been uniformly revised to “rams”.
- Line 332: It “Cloete”.
- Line 341: The space has been added.
- Line 352: The author's name has been changed into Antonini.
Thank you and best regards.
Yours sincerely,
Ruijun Wang
Inner Mongolia Agricultural University
306 Zhaowuda Road, Saihan District, Hohhot, P.R.China, 010018
Mobile: 086-15648116152
E-mail: imauwrj@163.com

Round 2
Reviewer 1 Report
Comments and Suggestions for Authors
Once revised the version 2 of this manuscript, in my opinion the introduction has been essentially improved, as well as the material and method section; now, statistical models have been adequately described. Tables now are referred in the text being headers and captions full informative.
However, some issues persist yet:
-The authors claim that they have modified the names of males, females and juvenile goats throughout the text, but this is not the case. Please check this.
- In table 1, units (kg) are written in the header of the Mean column; but there are several variables with different units. One solution could be write units in horizontal lines, following the name of each variable.
- In discussion, the previously described problems also persist. Most of references are about sheep, but the present work is about goat; please justify this situation and clearly indicate the species to which each reference refers. Also, in discussion is very important using the adequate names for males, females and juvenile goats.
- References have been revised but several references seem incomplete yet: please check.
- English language has been check and improved, but an error has been detected in the title: “Trraits” must be replaced by “traits”

Author Response
Dear reviewers and editors
I have modified the goat sex names in the whole paper according to your suggestions, the trait units in Table 1 have been added, and the "Trraits" in the title has been changed to "Traits". The breeds mentioned in the references in the paper have been indicated.
Although this study was carried out on goats and some references were on sheep, the research method is the same, so it can be used for reference.
If it still has any problem with my article, please contact me by E-mail.
I am deeply sorry for the inconvenience caused to you.
Thank you and best regards.
Yours sincerely,
Ruijun Wang
Inner Mongolia Agricultural University
306 Zhaowuda Road, Saihan District, Hohhot, P.R.China, 010018
Mobile: 086-15648116152
E-mail: nmgwrj@126.com